# Inulin as a Biopolymer; Chemical Structure, Anticancer Effects, Nutraceutical Potential and Industrial Applications: A Comprehensive Review

**DOI:** 10.3390/polym17030412

**Published:** 2025-02-04

**Authors:** Isaac Karimi, Mahnaz Ghowsi, Layth Jasim Mohammed, Zohreh Haidari, Kosar Nazari, Helgi B. Schiöth

**Affiliations:** 1Research Group of Bioengineering and Biotechnology, Laboratory for Computational Physiology, Department of Biology, Faculty of Science, Razi University, P.O. Box 67149-67346, Kermanshah, Iran; z.haidari@stu.razi.ac.ir (Z.H.); kosarnazari@stu.razi.ac.ir (K.N.); 2Department of Biology, Faculty of Science, Razi University, P.O. Box 67149-67346, Kermanshah, Iran; ghowsi.mahnaz@razi.ac.ir; 3Department of Medical Microbiology, College of Medicine, Babylon University, Hilla City 51002, Babylon Governorate, Iraq; med996.layth.jasim@uobabylon.edu.iq; 4Department of Surgical Sciences, Functional Pharmacology and Neuroscience, Uppsala University, 751 24 Uppsala, Sweden

**Keywords:** fructans, inulin, prebiotics, *Bifidobacterium*, pharmaceutics, nutraceuticals

## Abstract

Inulin is a versatile biopolymer that is non-digestible in the upper alimentary tract and acts as a bifidogenic prebiotic which selectively promotes gut health and modulates gut–organ axes through short-chain fatty acids and possibly yet-to-be-known interactions. Inulin usage as a fiber ingredient in food has been approved by the FDA since June 2018 and it is predicted that the universal inulin market demand will skyrocket in the near future because of its novel applications in health and diseases. This comprehensive review outlines the known applications of inulin in various disciplines ranging from medicine to industry, covering its benefits in gut health and diseases, metabolism, drug delivery, therapeutic pharmacology, nutrition, and the prebiotics industry. Furthermore, this review acknowledges the attention of researchers to knowledge gaps regarding the usages of inulin as a key modulator in the gut–organ axes.

## 1. Introduction

Valentin Rose the Younger, a German pharmacologist, isolated inulin, a fructan, from the roots of *Inula helenium* in 1804, and Thomson coined the term inulin in 1917 (for a review see [1]. Inulin is the second most abundant carbohydrate storage in plants and is distributed across various parts such as bulbs, roots, root tubers, leaf bases, grains, and fruits. Dicotyledonous plants such as the Asteraceae and Campanulaceae families are rich sources of inulin. Nutritional inulin-containing plants include leek, onion, garlic, asparagus, Jerusalem artichoke, dahlia, chicory, yacon, etc. (Figure 1; [2]). Therefore, inulins with various origins are found in nature, and inulin and its inulin-laden products have been used in different applications, mainly in the food and drug industries, in all countries for more than two centuries.

Inulin is not enzymatically assimilated in the upper alimentary tract; therefore, it does not alter blood glucose levels nor interfere with the insulin–glucagon counter-balance [3]. Basically, inulin is a prebiotic, which is a non-digestible ingredient resistant to gastric acidity and enzymatic digestion. At the same time, inulin can be fermented by the colonic microbiota and can stimulate the growth and activity of some colonic beneficial bacteria necessary for health and well-being. The inulin-type fructans (inulin and fructooligosaccharides (FOS)) have prebiotic effects. Because inulin has β (2→1) linkages, it reaches the colon intact, without enzymatic hydrolysis in the upper gastrointestinal tract. It is hydrolyzed by β-fructosidase-producing bacteria and increases the *Bifidobacteria* population (bifidogenic effect) in the colon [4]. Therefore, the demand for inulin-based products will skyrocket in the next decade mainly because of their usage as prebiotics in health and diseases (www.transparencymarketresearch.com; accessed on 20 December 2024; [5]).

## 2. Methodology of Literature Review

An electronic literature search of various databases was stochastically performed for publications in English. These databases included PubMed, Web of sciences, Embase, and Scopus. This stochastic review recaps the chemical characteristics and various applications of inulin.

## 3. Chemical Structure

In the inulin structure, linear chains of fructosyl groups are linked by (2→1) glycosidic bonds and α-D-(1→2) glucopyranoside rings are at their ends (Figure 2; Table 1). Plant-derived inulins have 2–100 fructose units and their aqueous solubility depends on chain length (Figure 2; [6]). Usually, inulin has a linear structure but some inulins have branches connected by β-(2→6) linkages. Based on their degree of polymerization (DP), inulin-type fructans can be divided into FOS (DP ≤ 10) and inulin (DP > 10; [7]). The DP of plant-derived inulin varies from 2–60 and its molecular weight is less than 10^4^ Da; however, microbial inulin has a higher molecular weight of >10^6^ Da. Nonetheless, it is not completely clear whether their physiological properties are different or not [8]; however, various FOS compounds (*n* = 13) and FOS substances (*n* = 113) were reported with their DP in PubChem (https://pubchem.ncbi.nlm.nih.gov/ accessed on 12 December 2024; Appendix A). It seems that the physicochemical properties of various FOSs and inulins are different and dependent on DP. In essence, the molecular diversity of FOS and inulin enables investigation into their quantitative structure-activity relationship for the discovery of novel variants in both plant and microbial resources.

The molecular weight of short-chain inulin depends on its chain length and varies between 1500–1600 Da [2]. For fructan production in plants, sucrose–sucrose fructosyltransferase (EC 2.4.1.99) catalyzes the transfer of a fructose molecule from one sucrose moiety to another for the formation of a kestose known as glucosyl-1, 2 fructosyl-1, 2 fructose. Fructan–fructan–fructosyltransferase (EC 2.4.1.100) elongates the inulin chain [10]. In addition, inulin can be produced by some microorganisms via a fructosyltransferase called inulosucrase (EC 2.4.1.9, sucrose: 2, 1-β-D-fructan 1-β-D-fructosyltransferase). This enzyme transfers fructose residues from sucrose and during this process, a polysaccharide chain connected by β-(2, 1) fructosyl linkages is formed and inulin polymers are generated [11]. Interestingly, one study performed enzymatic production of inulin. The recombinant *Lactobacillus gasseri* inulosucrase produced inulin from sucrose as the sole substrate with an average molecular weight of 5.858 × 10^6^ Da under the optimal conditions of pH 5.5, 25 °C, an enzyme dosage of 4.5 U/g sucrose, a sucrose concentration of 50% (*w*/*v*), and a reaction time of 1.5 h [12]. In addition, neo-inulin and neo-levan types of fructans with internal glucose residues can also be found [13]. Interestingly, another study [14] demonstrated that furanose-based polysaccharides, such as inulin, levan, and arabinan, exhibit a variety of conformational states. At the level of mono-, di-, or polysaccharides, the dynamic geometries of both the glycosidic linkages and the furanose rings remain largely unchanged and independent of one another. In summary, because of increased demands for inulin with different physicochemical and pharmaceutical features (for a review see [15], more tools in bioengineering and biotechnology must be employed to produce various inulin biopolymers or bio-oligomers with novel applications.

Despite the high market demand for inulin and inulin-based products, inulin faces several limitations in industrial applications, including its temperature sensitivity, which leads to degradation at high temperatures, restricting its use in processes like baking [16,17]. Its low solubility in water limits its functionality in certain beverages and syrups, while its variability in molecular weight can result in inconsistent properties, necessitating additional processing [18]. High doses of inulin may cause digestive discomfort, reducing its suitability for sensitive consumers, and its extraction and purification can be costly, making it less competitive compared to other fibers [19]. Furthermore, its functional benefits are concentration-dependent, and it is prone to hydrolysis in acidic conditions, limiting its stability in low-pH products. Limited consumer awareness and regulatory challenges in some regions also hinder its broader adoption. Addressing these issues through improved technologies, formulations, and education can help expand inulin’s industrial potential.

## 4. Pharmaceutical and Pharmacological Applications of Inulin

The LADMET (liberation, absorption, distribution, metabolism, excretion, and toxicity) of beta-D-Fructofuranose, FOS, and inulin were discussed in a review [15], which highlighted the liberation of drugs from various inulin-based drug delivery systems (DDSs) and the findings are partly presented in Appendix A. Inulin has many pharmaceutical applications (Figure 3). In this context, inulin can be used orally to deliver drugs to the colon. Physicochemically, inulin has a high molecular weight and melting temperature, low solubility, and it is slightly viscous when dissolved. Inulin has high molecular flexibility and low glass transition (Tg) in comparison with other oligo- or polysaccharides because of its (2→1) linked d-fructosyl backbone [20]. Because inulin has a low number of reducing groups, it is a suitable excipient for drugs, although the reducing groups can cause protein degradation. Inulin has low hygroscopicity and its crystallization rate is low. The aforementioned features make inulin a good stabilizer for proteins [21]. In this context, some studies have also shown that similarly to small sugars such as trehalose, mannitol and so on, inulin may be a good stabilizer for some pharmaceutical systems against physical and chemical degradation. For instance, one study showed that lyophilization of alkaline phosphatase with inulin preserved its activity [22]. Moreover, inulin renders a protective property to bovine plasma protein against denaturation during the freeze-drying procedure [23,24]. Inulin can be used for the spray freeze-drying of acyl-homoserine-lactone (AHL) acylases without loss of their activities [21]. This may be due to high molecular flexibility and because of the low steric hindrance of inulin [21]. In this regard, it has been shown that inulin protects PEGylated lipoplexes against aggregation during lyophilization and storage for three months. Likewise, inulin can be an appropriate stabilizer for different PEGylated nanoparticles [21]. One study suggested that inulin preserves doxorubicin-loaded (PEG) 3-PLA nanopolymers during lyophilization [25]. Inulin sugar glasses conserved the activity and structural integrity of influenza virosomes during freeze-drying and storage [26]. In this vein, inulin can stabilize hemagglutinin during freezing and freeze-drying and it is a suitable cryo-and lyo-protectant for this subunit of influenza vaccine [27]. The inclusion of Δ^9^-tetrahydrocanabidol (THC) as an unstable drug in inulin by spray-freeze drying and lyophilization strongly increased its stability [28,29]. To sum up, the genuine aforementioned features of inulin can be bio-mimicked for use in various industries ranging from fabricating sugar glasses to coating enzymes and therapeutic peptides.

One of the most promising methods for the improvement of the oral bioavailability of poorly water-soluble drugs is solid dispersion using compounds like inulin. In the gastrointestinal tract, these drugs dissolve slowly and their bioavailability is low. Inulin exhibits high solubility, particularly in water at elevated temperatures, which facilitates faster hydration and improved dissolution of poorly soluble drugs. This property makes it a valuable excipient in pharmaceutical formulations for enhancing drug bioavailability [21]. In this regard, one study compared the applicability of inulin, one surface-active derivative of inulin called inutec R Sp1, and polyvinylpyrolodone (PVP), as carriers in solid dispersions prepared by spray-freeze drying [30]. These carriers served to enhance the solubility of some lipophilic drugs such as diazepam, fenofibrate, ritonavir, efavirenz. The results showed that the dissolution rate of these drugs in the solution of various drugs was raised in the following order: inulin 2.3 kDa < PVP k30 << inutec R SP1. Another study showed that the incorporation of super disintegrants in solid dispersion tablets that contain the drug enhanced the solubility level of fenofibrate [31]. Solid dispersion tablets composed of inulin 4 kDa showed a fast dissolution capability and excellent storage stability [31]. A fast-dissolving, inulin-based solid dispersion tablet containing poorly soluble human immunodeficiency virus protease inhibitor, TMC240, has been marketed [32]. Research findings showed that solid dispersion of TMC240 in an inulin matrix amplified the in vitro and in vivo release of TMC2400. Different amorphous inulin types can be used as filler binders for tablets prepared by direct compaction and inulins with longer chains that have slower dissolution can be used as filler binders for chewable tablets or lozenges [21]. One study investigated the dissolution behavior of tablets composed of poorly aqueous soluble diazepam and inulin DP11 and inulin DP23. In this case, diazepam was incorporated in these inulin types in the amorphous state by freeze-drying using solvents like water and tertiary butyl alcohol. The study showed that fast dissolution of diazepam was observed with slow-dissolving carriers, such as these types of inulins [28]. In another study, labile lipophilic THC molecules were incorporated in a glassy matrix of inulin and the authors showed that this inulin-based solid dispersion may be an approach to fabricating tablets of THC for sublingual administration because this formulation released the THC very fast—in around 3 min [33]. Therefore, inulin-based biopolymers can enhance the bioavailability of an array of drugs that have low human intestinal absorption and may improve the LADME criteria of drugs.

In addition to the aforementioned role of inulin as a stabilizer, it was employed as a drug carrier for various organs [15,16,17,18,19,20,21,22,23,24,25,26,27,28,29,30,31,32,33,34]. Routinely, inulin has been used as the “gold standard” for measuring glomerular filtration rate (GFR) because after intravenous administration, inulin does not bind to the blood proteins and after its filtration by the kidneys, it is not reabsorbed, metabolized or secreted by kidneys. These properties make inulin a unique material for measuring GFR [9] and an effective drug carrier for nephron-related pathologies. Accordingly, potential DDSs were manufactured, which can be employed for the direct delivery of anti-infective or anti-inflammatory drugs straight to the urinary tract [9]. Moreover, inulin can be used to form a self-assembled micelle system in water which incorporates vitamin E (hydrophobic drug) and releases it in a well-ordered manner. For instance, vitamin E-succinate was linked to inulin by an ester bond and formed amphiphilic inulin-based polymers with hydrolyzable groups. In this context, particulates of hydroxyapatite inulin incorporated in novel poly(e-caprolactone) (PCL) microporous matrices have been used to produce bone substitutes and for the precise delivery of bioactive macromolecules that may be used in formulating long-term, controlled release devices for bioactive molecules such as hormones and extended-residence supports for tissue development and cell growth [35]. One of the most important challenges in medical research is targeting drugs to the large intestine for treating colonic disorders. Free films were prepared using a combination of inulin as a bacterially degradable system and time- or pH-dependent polymers as a coating formulation for colonic DDSs [36]. The utilization of inulin as a preparatory polymer starting material for the production of DDSs is based on its unique properties. For example, it is degradable by colon bacteria and it and its metabolites are nontoxic [35]. Inulin was also mixed with divinyl sulfone and succinic anhydride to formulate a biodegradable hydrogel through crosslinking with trimethylolpropane tris (3-mercaptopropionate). The resulting hydrogels were saturated with the 2-anti-cancer model drug methoxyestradiol (2-ME), an endogenous angiogenic metabolite of estrogen, and evaluated for their therapeutic potential in colorectal carcinoma and in vitro release. These pH-sensitive hydrogels had extraordinary properties such as swelling, stability in virtual gastric fluid, and biodegradability in the presence of inulinase and esterase. These worthy features make hydrogels suitable candidates for treating colorectal cancer through their efficient drug delivery [37,38]. To prepare amphiphilic inulin graft copolymer that can self-assemble in water into a micelle-type structure and to deliver the anticancer model drug doxorubicin, inulin was chemically modified and conjugated with polyethylene glycol 2000 and succinyl-ceramide [39]. These polymeric micelles released intact doxorubicin for a sustained period without a first burst release. An inulin-based glycovesicle was reported for pathogen-targeted drug delivery to control salmonellosis [40]. In summary, these are a few examples of the many applications of inulin in modern pharmaceutical technology and DDSs.

Inulin can also be applied in pharmaceutics as an auxiliary therapeutic agent for certain diseases (for a review see [21]. In this context, inulin-type fructans hasten the absorption of some minerals such as calcium, magnesium, copper, iron, and zinc in both humans and animals [41,42]. Broadly, various studies show that inulin can increase the calcium absorption and deposition of calcium in the bone of rats and humans (for a review see [43] and may be helpful in the prevention of calcium-deficient conditions like osteoporosis and rapid somatic growth. Similarly, the administration of inulin-type fructans increased the absorption of minerals and bone mineral accretion when combined with probiotic *Lactobacilli*. Inulin may increase absorption surface and expression of calcium-binding proteins in the large intestine and decrease intestinal lumen pH which in turn, increases the solubility of minerals in the gut. Furthermore, inulin suppresses bone resorption and increases the bioavailability of phytoestrogens by stimulating beneficial microorganisms in the gut. In addition, ITFs can improve mineral absorption by stabilizing the intestinal flora and reducing inflammation. Therefore, its consumption may be beneficial in the prevention and treatment of musculoskeletal diseases [41,42,43,44]. Consistently with the aforementioned studies, consumption of short- and long-chain inulin fructans (8 g/day for one year) in young adolescents increased calcium absorption and bone mineral density [45]. Butyrate is the main fermentative product of inulin and is a strong stimulator of CaBP-9kDa expression that stimulates the calcium absorption pathway through increasing 1, 25 (OH)2D3 receptor binding [46]. Similarly, two studies reported that 75 g/kg inulin in a semi-purified diet stimulated the absorption of magnesium, copper, and zinc in rats of different ages [47,48]. In this context, young male rats treated with inulin-type fructans increased the phosphorus content of bone in two weeks [49]. Various factors such as the dietary content of minerals or fructans and background diet, sex, age, and hormonal status may alter the effects of inulin-type fructans on the absorption of minerals [41]. In essence, the mechanisms of improved intestinal absorption of minerals in the presence of inulin-based formulation need to be investigated more deeply. Technically, various industrial efforts are welcomed to investigate the fermentative products of inulin using different bioreactors to develop formulations for the better absorption of minerals.

Inulin has many therapeutic potentials in metabolic and infectious diseases. For instance, high doses of inulin (40–100 g/d) are effective in diabetic patients [50]. The dietary supplementation of 8 g of FOS for two weeks reduced serum glucose levels in type II diabetic patients, while dietary supplementation with FOS did not alter serum glucose in healthy subjects [51]. As mentioned above, inulin has (2–1) linkages and human digestive enzymes such as α-glucosidase, maltase, isomaltase, and sucrase cannot hydrolyze these linkages; therefore, pristine inulin will be fermented and converted to lactate, gases, and short-chain fatty acids (SCFAs) such as acetate, propionate, and butyrate in ceco-colon [10,11,12,13,14,15,16,17,18,19,20,21,22,23,24,25,26,27,28,29,30,31,32,33,34,35,36,37,38,39,40,41,42,43,44,45,46,47,48,49,50,51,52]. Perhaps the hypoglycemic effects of inulin may be due to propionate production because it exerts a hypoglycemic effect via inhibition of hepatic gluconeogenesis [26,27,28,29,30,31,32,33,34,35,36,37,38,39,40,41,42,43,44,45,46,47,48,49,50,51,52]. Inulin decreases blood levels of fatty acids and triacylglycerols and their hepatic biosynthesis. An intake of 8 g of short-chain FOS/d for two weeks decreased fasting blood glucose and serum total cholesterol levels in diabetic patients. Nonetheless, serum high-density-lipoprotein (HDL) cholesterol, triacylglycerols, and fatty acids were not significantly altered. The mechanisms by which short-chain FOS can modulate glucose and lipid metabolism are not completely known [53]. Moreover, some animal and human studies have demonstrated that inulin and oligofructoses obtained from inulin have lipid-lowering effects [54]. In rats, long-term consumption of high doses of oligofructoses reduced blood cholesterol levels [54,55,56]. It has been reported that inulin consumption in breakfast cereal (9 g/d) or as a powdered addition to beverages and meals (10 g/d) reduced fasting triacylglycerols up to 27 and 19%, respectively [54]. These effects may be mediated by reducing the secretion of very low-density lipoprotein (VLDL) particles from the liver and reducing the activity and gene expression of fatty acid synthetase [54]. One study hitherto showed that consumption of 14 g/day inulin for 4 weeks did not affect the fasting total, low-density lipoprotein (LDL) or HDL cholesterol, or serum triacylglycerols [57]. Likewise, the modest hyperlipidemic subjects taking inulin showed reduced LDL levels without concomitant alteration in HDL cholesterol and serum triacylglycerols levels [58]. Inulin may be useful in regulating nitrogen balance by reducing blood urea and uric acid concentration [10]. In an impressive review [59] the lipid-lowering properties of dietary polysaccharides like inulin have been discussed.

The exact therapeutic and prophylactic effects of inulin for metabolic syndrome have not been determined, but an array of reviews have focused on the improvement of some of the therapeutic effects of inulin in this case [60,61,62,63,64]. These systematic reviews pertained to the microbial community of the gut and its roles in stabilizing the gut barrier and producing useful postbiotics that modulate some receptors that are involved in glucose and fatty acid metabolism. To the best of our knowledge, the direct interaction of inulins and FOSs with the mucous membrane of the gut has not been discussed (Figure 4).

Inulin may be useful in regulating nitrogen balance by reducing blood urea and uric acid concentrations [10]. Despite the many reviews (e.g., [65,66]) that report on the metabolic effects of inulin in health and diseases, there is no study that pertains to the impact of inulin on amino acid and nucleotide intermediary metabolism. In conclusion, the mechanisms by which inulin and inulin-derived oligofructoses, and short-chain FOS can modulate glucose, amino acid, and fatty acid metabolism are mysterious and warrant future investigations.

## 5. Anticancer Effects of Inulin

Cancer is a leading cause of death, with a profound impact on healthcare systems worldwide. Challenges in prophylaxis and oncotherapy include chemotherapy resistance, off-target effects of chemotherapeutic agents that exhibit severe adverse effects, and the expensiveness of current chemotherapeutic agents. In recent years, inulin as a naturally occurring prebiotic fiber has gained ample attention for its potential in cancer treatment owing to its unique structure, stability, and nutritional properties which highlight it as an effective auxiliary therapeutic, adjuvant and carrier for drug delivery in cancer therapy (for a review see [34,67,68,69,70]; Figure 5). In this vein, efforts towards the development of therapeutic inulin-based nanomaterials and nanocomposites with a special emphasis on the multifunctional role of inulin in oncotherapy as a signaling, synergistic, immunomodulatory and anticarcinogenic molecule have been pursued [68,69,70]. A recent review [70], provides a succinct overview of current clinical trials and observational studies associated with inulin-based oncotherapy. Over the past decades, natural polysaccharides, as well as biopolymers, have been widely developed for targeting colon cancer using various DDSs [68]. Inulin was used in colorectal tumors not only as an entrapment material concerning its degradation by enzymes in the colonic microflora and its drug release behavior in a sustained and controlled manner, but also as a dietary fiber with added well-being benefits. In conclusion, to the best of our knowledge, mechanisms beyond the anti-cancer effects of inulin have not been addressed until now.

## 6. Nutraceutical Potentials of Inulin

Several studies have evaluated the antioxidative activity of inulin (for a review see [71]). In this regard, one study showed that dietary supplementation with inulin improved the feed consumption and egg production rate of laying hens and increased the antioxidative activities of superoxide dismutase (SOD), catalase (CAT), glutathione peroxidase (GSH-Px), and serum total antioxidant capacity and reduced levels of malondialdehyde, an index of lipid peroxidation [72]. It has been reported that dietary supplementation of inulin increased *Bifidobacteria* and *Lactobacilli* counts in the ceca of laying hens. *Bifidobacterium* spp. are lactic acid-producing bacteria that have SOD and lactic acid, both of which help scavenge free radicals. Another study indicated that pre-treatment with inulin prevents impairment of the colonic smooth muscle cell contractility induced by lipopolysaccharide exposure by a decrease in mucosal production of free radicals [73]. Another study focused on the antioxidative activity of inulin-based prebiotics, which promotes the development of functional fermented goat milk, indicated that the radical scavenging rate reached 75.52% and the scavenging rate of superoxide radicals was 21.09%. The researchers pointed out that inulin improved the nutritional value of functional foods [74]. In a similar vein, an impressive and seminal study reported on the consumption of a combination of the probiotic bacteria *Lactobacillus casei* (4 × 10^8^ colony-forming unit (CFU)) and prebiotic inulin (400 mg) and suggested that dietary supplementation with this symbiotic may prevent oxidative stress in the plasma of healthy volunteers by increasing the ferric reducing ability of plasma (FRAP) and CAT activity [75]. In another study, the antioxidant activity of carboxymethyl inulin (CMI) was enhanced by chemical modification. The results revealed a significant enhancement in antioxidant activity upon the introduction of cationic Schiff bases into CMI as compared to the commercially available antioxidant Vc [76]. Therefore, the direct and indirect antioxidative effects of inulin that are mediated through the improvement of gut microbiota place inulin in a superior position to combat oxidative stress in both health and disease.

The regular intake of dietary fiber boosts the gut microbiome and health of the host in several ways. Dietary fibers (DFs) are indigestible products that have become a vital ingredient to be included in every healthy diet. DF is defined as carbohydrate polymers containing ≥ 10 monomeric units that resist digestion by endogenous enzymes in the small intestine. DF includes edible carbohydrate polymers, and synthesized carbohydrate polymers [65]. DF can be divided into “soluble DF” (SDF) and “insoluble DF” (IDF) according to solubility, and it can be further categorized into “partially fermentable fiber” and “completely fermentable fiber” based on its fermentability [65]. The health benefits of DF for hosts are revealed mainly by changes in gut microbiota composition and microbial metabolites. In this context, the role of inulin as a prebiotic DF is at the heart of the biopharmaceutical applications of this biopolymer, which may be due to its ability to modulate gut microbiota, enhance gut health and gut–organ axes (e.g., gut–kidney axis; Figure 6), and improve metabolic processes (for a review see [5,6,7,8,9,10,11,12,13,14,15]). Prebiotics can be described as “selectively fermented ingredients that alter the configuration and activity in the gastrointestinal microbiota that confer positive effect”. In a seminal review [77], the beneficial impact of the dietary inclusion of inulin in human and animal models was discussed. The authors concluded that inulin as a fructan prebiotic can be a part of functional food products that promote health benefits to consumers. The review also vindicates the efficacy of inulin as a stabilizer, fat replacer, component in nanoformulations, and humectant in the cosmetic industry. Interestingly, hydrophobically modified inulin (HMI) has applications like the targeted release of drugs. In essence, it is important to explore the properties of SCFA inulin esters because they are less studied. Furthermore, HMI stabilizes various dispersion formulations as an excipient for producing hydrophobic drugs because inulin is generally recognized as safe (Figure 7; [78]). Therefore, inulin and inulin-based products may have health-promoting activities such as acting as DF, prebiotics, and drug carriers in the gut.

The prebiotic impact of inulin on the management of gastrointestinal disorder has been the subject of many studies (for a review see [79]). These researchers reviewed the positive impact of prebiotics in experimental colitis and human inflammatory bowel disease (IBD), and concluded that inulin has trophic effects on gut microflora via the enhancement of colonic production of short-chain fatty acids (SCFAs) and the subsequent growth of indigenous lactobacilli and bifidobacteria, decreasing mucosal lesion and mucosal inflammation by promoting host defense against invasion and pathogen translocation, and inhibiting gastrointestinal diseases like IBD. The therapeutic effects of prebiotic inulin-type fructans in severe bowel diseases like active and inactive Crohn’s disease need more deep investigations [80]. In another review [19], the history of inulin as a prebiotic and its ability to enhance the growth and functionality of Bifidobacterium bacteria, as well as its effect on host gene expression and metabolism, were discussed. Finally, the authors proposed symbiotic (inulin plus probiotics) applications in the management of a group of diseases including cardiometabolic diseases after a deep discussion about various controversies regarding the effects of inulin on the health and diseases of the gut. In this continuum, the symbiotic formulations of inulin and other probiotics were reviewed [19,20,21,22,23,24,25,26,27,28,29,30,31,32,33,34,35,36,37,38,39,40,41,42,43,44,45,46,47,48,49,50,51,52,53,54,55,56,57,58,59,60,61,62,63,64,65,66,67,68,69,70,71,72,73,74,75,76,77,78,79,80,81]. Moreover, a bibliographic review focused on human clinical studies highlighted the main effects of inulin on human metabolic health, with a special focus on the mechanisms of action of this prebiotic. The authors concluded that inulin supplementation contributes to anthropometric indices control and improves metabolic status mainly through the selective favoring of SCFA-producer species from the genera Bifidobacterium and Anaerostipes [66]. In a seminal review [82], the role of inulin in the establishment of the gut microbial community during the first 1000 days of a child’s life was discussed, and the impact of inulin on the prevention of enteric diseases in adulthood was highlighted, in addition to the role of fructans in metabolic programming. In this context, another review [83] pertained to the microbial-derived products, including SCFAs, lipopolysaccharides and secondary bile acids, that may be involved in the regulation of hepatic lipid metabolism. Finally, they concluded that the relationships between bacterial species (e.g., competition and mutualism), play key roles in the degradation of inulin and the regulation of the microbial structure. Regarding the therapeutic effects of inulin-type fructans (ITFs) and prebiotics on IBDs and Crohn’s disease, the findings of different studies are controversial, possibly due to the altering microbial community (for a review see [84,85]. In a systematic review [86], all aspects of inulin-type fructans including short-chain fructooligosaccharides (scFOS), oligofructose, and inulin were deeply discussed and the authors proposed the personalization of prebiotic applications in precise and personalized medicine. To personalize the ITFs, we must focus on the plasticity of the microbial community of the intestine, postbiotics produced during microbial modulation, and their effects on extra-intestinal tissues (for a review, see [87,88]. In addition to the DF formulation of inulin, innovative synthetic or semi-synthetic inulin-based delivery systems, such as hydrogels and nanoparticles, are designed for their sustained and controlled-release formulations. The mechanisms and bacterial enzymes involved in inulin degradation by gut microbiota have been reviewed [89]. An inulin-rich flour has been formulated from *Smallanthus sonchifolius*, popularly known as yacon, which is a member of the *Asteraceae* family (for a review see [64]. The authors showed that the intake of yacon flour can reduce glycemia, HbA1c, advanced glycation ends, plasma lipids, body fat mass, body weight, and waist circumference and improve intestinal microbiota and antioxidant status. In conclusion, inulin has many applications in the nutraceutical sector, especially for innovating new functional formulations in feed/food technologies; however, further studies are welcomed to assess its effects on gut–organ axes.

An array of investigations focused on the food and feed markets and the associated technologies of inulin and its byproducts (Figure 8) [7,20,21,34,71,81,90,91,92,93,94,95,96,97,98].

## 7. Conclusions and Remark

Inulin can be considered a treasure trove for many applications in various industries, and many countries are involved in mass producing this economic polysaccharide (https://www.fortunebusinessinsights.com/industry-reports/inulin-market-101512, accessed on 23 January 2025). In this overview, we have tried to consolidate the current and possible applications of inulin by reviewing seminal studies, current reviews, and systematic reviews focused on inulin to offer an integrated package, and to highlight the knowledge gaps. Nowadays, inulin and ITFs are predominantly used as prebiotics to engineer gut microbiota and alleviate metabolic diseases like diabetes mellitus; however, inulin’s role in modulating gut–organ axes remains to be investigated deeply. In addition, studies on the non-medical applications of inulin in synthesizing new bio-engineering products are scarce. Key areas for the usage of inulin products are the dairy, bakery, beverage, and meat industries as sugar and fat replacements, despite their shortcomings. The application of inulin as an immunosaccharide in the aquaculture sector would be a new opportunity that warrants more investigation. In conclusion, a quick search of inulin in Google images will clarify the many medicinal and industrial products made of inulin in the global market. This review has tried to demonstrate the economic importance of inulin in a comprehensive manner.

## Figures and Tables

**Figure 1 polymers-17-00412-f001:**
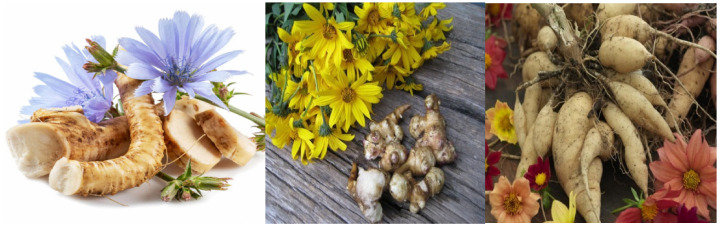
Well-known plant resources of inulin. From left to right, the root of chicory (*Cichorium intybus*), sunchoke (*Jerusalem artichoke*), and dahlia tubers (*Dahlia pinnata*).

**Figure 2 polymers-17-00412-f002:**
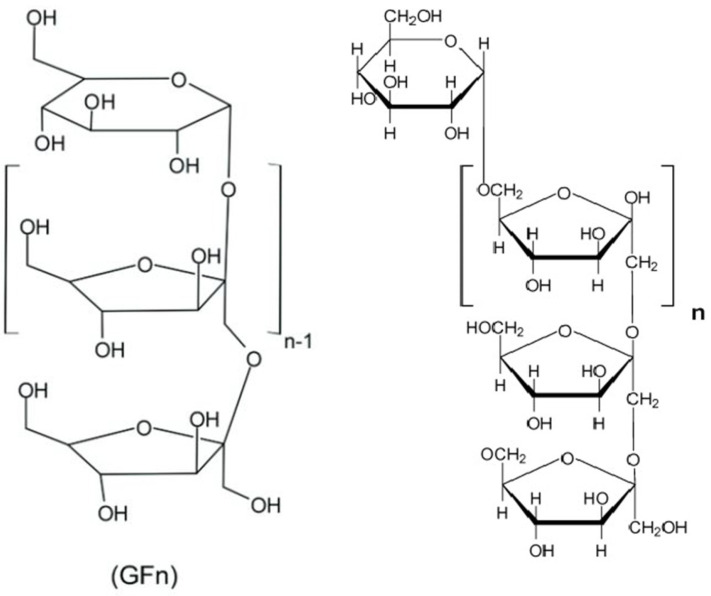
Inulin (C_6_H_10_O_5_)_n_ (**left**) and fructooligosaccharide (C_6_H_10_O_5_)_n_ (**right**) structure [9].

**Figure 3 polymers-17-00412-f003:**
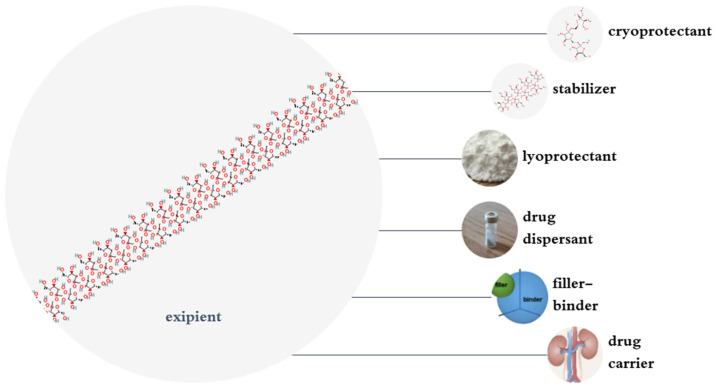
Various applications of inulin in the pharmaceutical industry.

**Figure 4 polymers-17-00412-f004:**
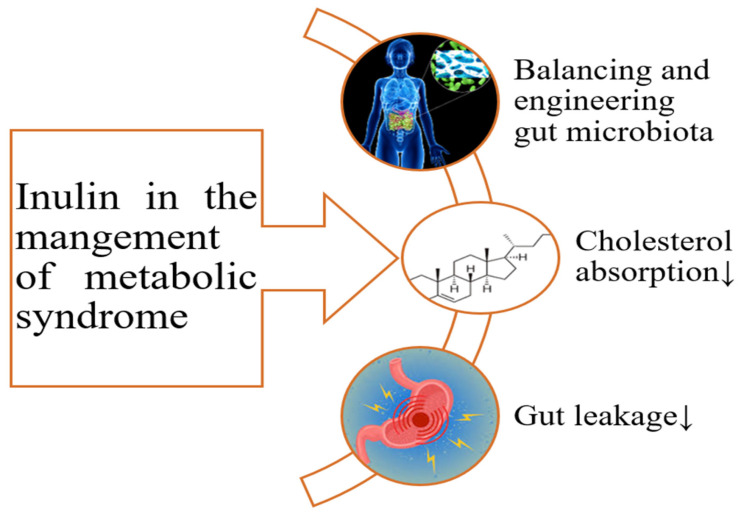
The impact of inulin on the management of metabolic syndrome; Note: ↓ shows decline.

**Figure 5 polymers-17-00412-f005:**
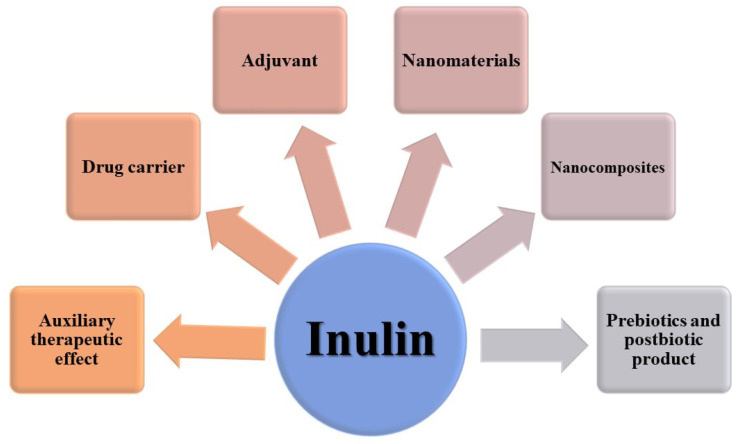
The various portals of inulin applications in oncotherapy.

**Figure 6 polymers-17-00412-f006:**
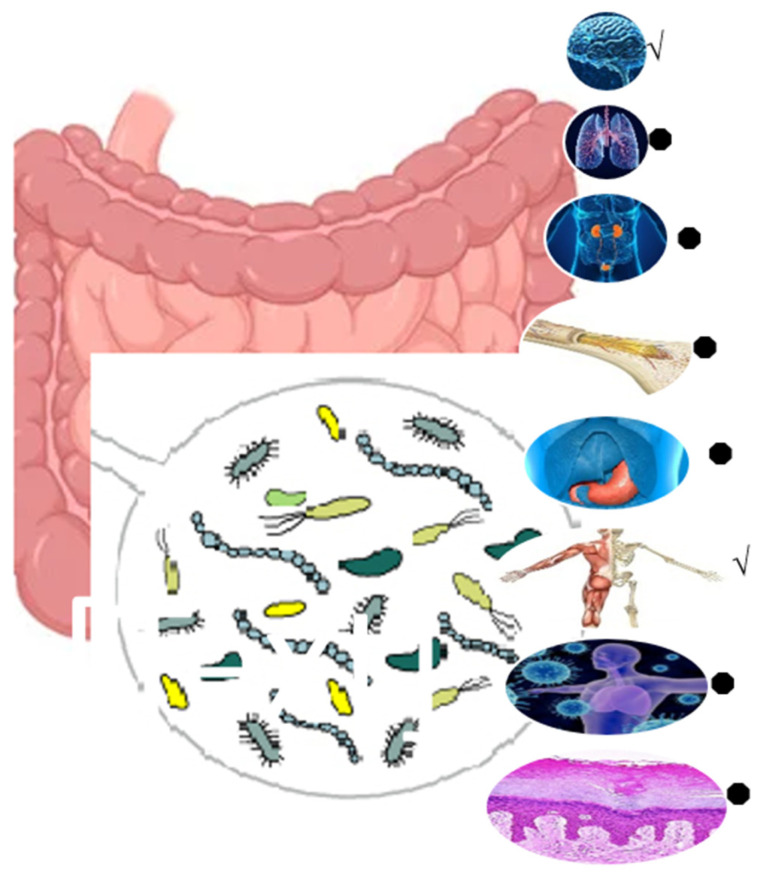
The effect of inulin on the gut–organ axes. √ and ● express axes with greater and fewer publications, respectively.

**Figure 7 polymers-17-00412-f007:**
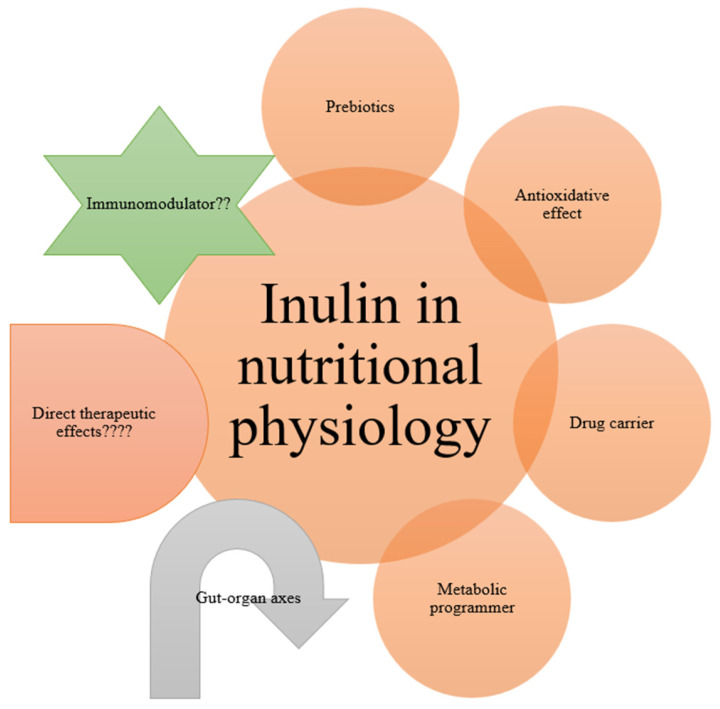
The role of inulin in nutritional physiology.

**Figure 8 polymers-17-00412-f008:**
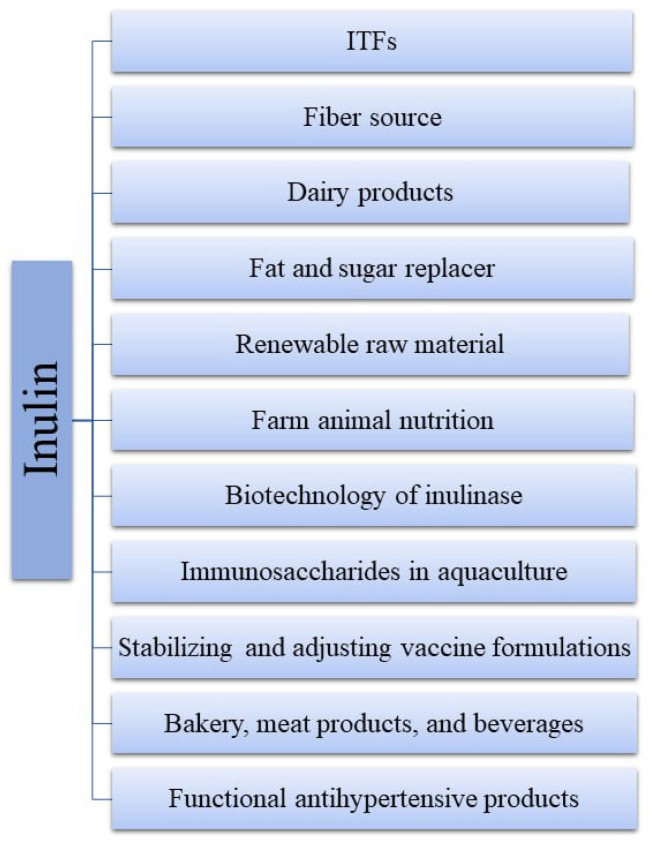
Schematic summary of possible applications for the global market of inulin. ITFs: inulin-type fructans.

**Table 1 polymers-17-00412-t001:** The physicochemical properties of inulin compounds reported by https://pubchem.ncbi.nlm.nih.gov/ accessed on 12 December 2024.

	Name/PubChem CID	Molecular Weight g/mol	XLogP3-AA	Hydrogen Bond Donor Count	Hydrogen Bond Acceptor Count	Rotatable Bond Count	Topological Polar Surface Area Å	Canonicalized
C_228_H_382_O_191_ 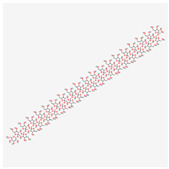	Inulin [USP: BAN]/18772499]	6179	−70.2	116	191	149	3040	Yes
C_24_H_42_O_21_ 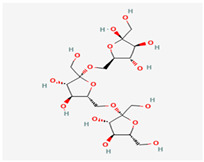	Inulin—chicory Inulin—*Jerusalem artichoke*/132932783	666.6	−7.4	14	21	13	384	Yes
C_18_H_32_O_16_ 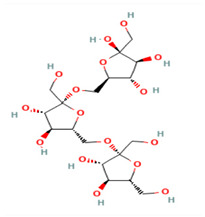	Levan—*Erwinia herbicola* 440946	504.4	−6	11	16	10	296	Yes

Note: Inulin; Linear chain of β(2→1)-linked fructosyl groups with α-D-(1→2) glucopyranoside at the end and Levan; linear or branched chain of β(2→6)-linked fructosyl groups, with possible β(2→1) branches.

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
