# Peer review of "Inulin as a Biopolymer; Chemical Structure, Anticancer Effects, Nutraceutical Potential and Industrial Applications: A Comprehensive Review"

_polymers, 2025, doi:10.3390/polym17030412_

Round 1
Reviewer 1 Report
Comments and Suggestions for Authors
The manuscript "Inulin as an Epoch-Making Economic Biopolymer for Diverse Industries: A Comprehensive Review" is devoted to a relevant topic, is well structured and contains elements of novelty. Authors analyzed a large amount of modern literature to write the text of the manuscript.
However, the review is not without a number of significant comments.
1. The Title of the review includes "Economic Biopolymer", however, the text of the article does not discuss the economic feasibility of using inulin in industry at all. The economic component is certainly important, so the authors should have devoted at least a small section of several paragraphs to it.
2. Section 3 should mention the dynamic conformational mobility of the inulin molecule, especially in comparison with other types of widespread polysaccharides (starch, cellulose and its derivatives, chitosan, etc.).
3. Since many reviews have already been written about the use of inulin in human economic activity, to which the authors themselves refer, then in order to increase the interest of readers, the authors should emphasize the differences in their review, i.e. more clearly highlight the scientific novelty. For example, it would be interesting to more clearly define the influence of the molecular weight of inulin on its beneficial properties and application prospects, i.e. compare the features of the use of inulin and fructooligosaccharides in industry. This material is already contained in fragments in the text of the manuscript. It should be summarized.
In addition (optionally), it would be possible to add a small Section on the differences in the prospects for the use of inulin in industry compared to other common polysaccharides with a wide range of biological activity and applied use – chitosan, carboxymethylcellulose, etc.
4. In order to make the review more analytical and increase the interest of readers, the authors should add an additional Section on the limitations of the use of inulin in industry and the difficulties in working with it for applied purposes.
5. To increase the scientific novelty of the manuscript, the authors should add a Section on past trends in the development of research on the prospects for the use of inulin in industry and what has changed in these trends in the last 10 years or so, how current trends differ from trends of ten years ago, what has been implemented and what has not been implemented and for what reasons. This Section will be a logical continuation of the Section on the limitations of the use of inulin in industry.
Technical shortcomings:
1. In Figure 2, the formulas of inulin and FOS are presented in different forms. They should be made uniform.
2. On Lines 79-80, the authors write that the molecular weight of inulin varies within 1500-1600 Da, however, on Line 90 the weight is 5,858,000, and in Table 1 it is 6179 Da. This contradiction should be eliminated.
3. It is not entirely clear from the text what the authors wanted to say in the second line of Table 1. Probably, these are α-D-(1→2) glucopyranoside rings at the end of the inulin chain, but then why does the table also include levan?
4. Supplementary file S2 is mentioned on Line 102, there is no link to Supplementary file S1 and only one Supplementary file is attached to the article.
5. Line 137: the authors write "Inulin has high solubility which helps faster hydration of poorly soluble drugs", this is indeed true, but at elevated temperatures. This point should be clarified.
6. The design of Figure 5, in my opinion, is not successful. Why did the authors make such a variety of frames?
7. The design of Figure 8 is also not successful. It should be changed so that the fonts in the Figure can be made larger.
8. The correctness of the References is questionable. I'm not sure that the authors' names should be written in capital letters.
Author Response
The manuscript "Inulin as an Epoch-Making Economic Biopolymer for Diverse Industries: A Comprehensive Review" is devoted to a relevant topic, is well structured and contains elements of novelty. Authors analyzed a large amount of modern literature to write the text of the manuscript.
However, the review is not without a number of significant comments.
- The Title of the review includes "Economic Biopolymer", however, the text of the article does not discuss the economic feasibility of using inulin in industry at all. The economic component is certainly important, so the authors should have devoted at least a small section of several paragraphs to it.
Response (R): We changed title into "Inulin as a biopolymer; Chemical structures, anticancer effects, nutraceutical potential and industrial applications: A comprehensive review” to show the content of manuscript and expand readership.
- Section 3 should mention the dynamic conformational mobility of the inulin molecule, especially in comparison with other types of widespread polysaccharides (starch, cellulose and its derivatives, chitosan, etc.).
R: Interestingly, in another study (Nester and Plazinski, 2020) demonstrated that furanose-based polysaccharides, such as inulin, levan, and arabinan, exhibit a variety of conformational states. At the level of mono-, di-, or polysaccharides, the dynamic geometries of both the glycosidic linkages and the furanose rings remain largely unchanged and independent of one another.
- Since many reviews have already been written about the use of inulin in human economic activity, to which the authors themselves refer, then in order to increase the interest of readers, the authors should emphasize the differences in their review, i.e. more clearly highlight the scientific novelty. For example, it would be interesting to more clearly define the influence of the molecular weight of inulin on its beneficial properties and application prospects, i.e. compare the features of the use of inulin and fructooligosaccharides in industry. This material is already contained in fragments in the text of the manuscript. It should be summarized.
R: After discussing with authors and the first comments we changed the title and in this regard, to connect molecular weight or other physicochemical attribute to the economic importance, we need try another work and here we decided to be more superficial.
In addition, (optionally), it would be possible to add a small Section on the differences in the prospects for the use of inulin in industry compared to other common polysaccharides with a wide range of biological activity and applied use – chitosan, carboxymethylcellulose, etc.
R: More interesting! but we cannot respond to this impressive question in this review, however it is our pleasure to follow this hypothesis in collaborating reviewer (if he/she wanted not to be anonymous) in another work.
- In order to make the review more analytical and increase the interest of readers, the authors should add an additional Section on the limitations of the use of inulin in industry and the difficulties in working with it for applied purposes.
R: We added a paragraph to respond this comment: Despite high demand of market for inulin and inulin-based products, inulin faces several limitations in industrial applications, including its temperature sensitivity, which leads to degradation at high temperatures, restricting its use in processes like baking (Böhm et al., 2005; Glibowski and Bukowska, 2011). Its low solubility in water limits its functionality in certain beverages and syrups, while the variability in molecular weight can result in inconsistent properties, necessitating additional processing (Du et al., 2023). High doses of inulin may cause digestive discomfort, reducing its suitability for sensitive consumers, and its extraction and purification can be costly, making it less competitive compared to other fibers (Tawfick et al., 2022). Furthermore, its functional benefits are concentration-dependent, and it is prone to hydrolysis in acidic conditions, limiting its stability in low-pH products. Limited consumer awareness and regulatory challenges in some regions also hinder its broader adoption. Addressing these issues through improved technologies, formulations, and education can help expand inulin's industrial potential.
- To increase the scientific novelty of the manuscript, the authors should add a Section on past trends in the development of research on the prospects for the use of inulin in industry and what has changed in these trends in the last 10 years or so, how current trends differ from trends of ten years ago, what has been implemented and what has not been implemented and for what reasons. This Section will be a logical continuation of the Section on the limitations of the use of inulin in industry.
R: Impressive comment. Unfortunately, we do not free access to the information of economic data like market and market. At future, we will focus on these trends.
Technical shortcomings:
- In Figure 2, the formulas of inulin and FOS are presented in different forms. They should be made uniform
R: Yes, consistency has been unified.
- On Lines 79-80, the authors write that the molecular weight of inulin varies within 1500-1600 Da, however, on Line 90 the weight is 5,858,000, and in Table 1 it is 6179 Da. This contradiction should be eliminated. R: It has been checked. However, inulin can exist in a wide range of chain lengths, with molecular weights typically reported from 3000 Da to over 6000 Da, depending on the degree of polymerization (n), source and the type of inulin (short-chain or long-chain). Clarification of Values for molecular weight:
- The value of 1,500–1,600 Da (Lines 79–80) likely refers to a very short-chain inulin or oligosaccharide.
- The value of 6,179 Da (Table 1) is consistent with mid-range inulin.
- High Molecular Weight Inulin: The value of 5.858 × 10⁶ Da is supported by the enzymatic production process described in Ni et al., 2018. Enzymatic synthesis can sometimes produce highly polymerized or aggregated forms of inulin, leading to such high molecular weights. https://www.sciencedirect.com/science/article/abs/pii/S0141813017337601
- It is not entirely clear from the text what the authors wanted to say in the second line of Table 1. Probably, these are α-D-(1→2) glucopyranoside rings at the end of the inulin chain, but then why does the table also include levan? R: It has been separated. the levan in a different row because it has a distinct structure compared to inulin. Inulin contains α-D-(1→2) glucopyranoside rings at the chain end.
- Supplementary file S2 is mentioned on Line 102, there is no link to Supplementary file S1 and only one Supplementary file is attached to the article. R:The journal will deposit supplementary file in special space and hyperlinked to paper.
- Line 137: the authors write "Inulin has high solubility which helps faster hydration of poorly soluble drugs", this is indeed true, but at elevated temperatures. This point should be clarified. R: t has been clarified in the revised manuscript
- The design of Figure 5, in my opinion, is not successful. Why did the authors make such a variety of frames? R: Thank you for your thoughtful feedback. We truly appreciate your insight, which has helped us improve the manuscript. Figure 5 has been revised as per your suggestion, and we hope the updated version meets your expectations.
- The design of Figure 8 is also not successful. It should be changed so that the fonts in the Figure can be made larger. R: Thank you for your helpful feedback. Figure 8 has been revised with larger fonts as suggested. We truly appreciate your input!
- The correctness of the References is questionable. I'm not sure that the authors' names should be written in capital letters. R: Thank you for your valuable feedback. The references have been carefully reviewed and corrected, with authors' names formatted appropriately.
Reviewer 2 Report
Comments and Suggestions for Authors
The work has been developed correctly. It is a useful source of collected information on the usefulness of inulin and its derivatives, especially in the pharmaceutical industry and preventive health care.
I have a few observations that could help improve readers' understanding of the work.
1. In the description of the method, it would be useful to know what search terms were searched, how many publications were found, and how they were properly selected for review
2. The pathway that the drug substance takes in the organism involves LADME (liberation, adsorption, distribution, metabolism and elimination). The text indicates many applications of inulin or inulin-based biopolymers in the modification of drug formulation, indicating that these polymers have a significant effect on the release of the active substance. At the same time, the conclusions state that inulin specifically affects ADME, leaving out “liberaton.” I think this is worth completing.
3. Research on the effects of short-chain FOS on glucose and lipid metabolism is currently popular. While I do not negate the use of the 1984 publication in the review, it is difficult to assume that there are not more recent studies that can be added in line 257
4. It seems to me that according to the WHO, still cardiovascular diseases are the most common cause of death. The entry in line 292 suggests that these are cancers. I think this needs to be corrected.
Author Response
The work has been developed correctly. It is a useful source of collected information on the usefulness of inulin and its derivatives, especially in the pharmaceutical industry and preventive health care.
I have a few observations that could help improve readers' understanding of the work.
- In the description of the method, it would be useful to know what search terms were searched, how many publications were found, and how they were properly selected for review
- We did not decide to do systematic review or umbrella review, however in any applications of inulin, we want to do analytic reviews at future, here we wanted to meet the scope of special issue.
- The pathway that the drug substance takes in the organism involves LADME (liberation, adsorption, distribution, metabolism and elimination). The text indicates many applications of inulin or inulin-based biopolymers in the modification of drug formulation, indicating that these polymers have a significant effect on the release of the active substance. At the same time, the conclusions state that inulin specifically affects ADME, leaving out “liberaton.” I think this is worth completing.
I SUGGESED
The findings demonstrate that inulin and inulin-based biopolymers significantly influence the LADME pathway, particularly affecting the liberation, absorption, distribution, metabolism, and elimination of drug substances. Their role in modifying drug formulations and controlling the release of active substances underscores their potential in optimizing drug delivery systems.
R: It has been done. A review has been published in this regard (Khan et al., 2024) that was included in our study.
- Research on the effects of short-chain FOS on glucose and lipid metabolism is currently popular. While I do not negate the use of the 1984 publication in the review, it is difficult to assume that there are not more recent studies that can be added in line 257
R: Following reference has been replaced.
Liu, F., Prabhakar, M., Ju, J., Long, H. and Zhou, H.W., 2017. Effect of inulin-type fructans on blood lipid profile and glucose level: a systematic review and meta-analysis of randomized controlled trials. European journal of clinical nutrition, 71(1), pp.9-20.
It seems to me that according to the WHO, still cardiovascular diseases are the most common cause of death. The entry in line 292 suggests that these are cancers. I think this needs to be corrected. R: Thank you for your feedback. The correction has been made as suggested.
Round 2
Reviewer 1 Report
Comments and Suggestions for Authors
The authors took into account most of my comments. The controversial points were clarified by them. The title of the article was replaced with a more correct one. I propose to accept the manuscript for publication.